# Identification of Quantitative Trait Loci (QTL) for Sucrose and Protein Content in Soybean Seed

**DOI:** 10.3390/plants13050650

**Published:** 2024-02-27

**Authors:** Daniel R. Jamison, Pengyin Chen, Navam S. Hettiarachchy, David M. Miller, Ehsan Shakiba

**Affiliations:** 1Department of Crop, Soil, and Environmental Sciences, University of Arkansas, Fayetteville, AR 72701, USA; dan.jamison@outlook.com (D.R.J.); dmmiller@uark.edu (D.M.M.); 2Department of Food Science, University of Arkansas, Fayetteville, AR 72704, USA; nhettiar@uark.edu

**Keywords:** sugar and protein content, QTL: quantitative trait loci, SNP: single nucleotide polymorphism, RILs: recombinant inbred lines

## Abstract

Protein and sugar content are important seed quality traits in soybean because they improve the value and sustainability of soy food and feed products. Thus, identifying Quantitative Trait Loci (QTL) for soybean seed protein and sugar content can benefit plant breeders and the soybean market by accelerating the breeding process via marker-assisted selection. For this study, a population of recombinant inbred lines (RILs) was developed from a cross between R08-3221 (high protein and low sucrose) and R07-2000 (high sucrose and low protein). Phenotypic data for protein content were taken from the F2:4 and F2:5 generations. The DA7250 NIR analyzer and HPLC instruments were used to analyze total seed protein and sucrose content. Genotypic data were generated using analysis via the SoySNP6k chip. A total of four QTLs were identified in this study. Two QTLs for protein content were located on chromosomes 11 and 20, and two QTLs associated with sucrose content were located on chromosomes 14 and. 11, the latter of which co-localized with detected QTLs for protein, explaining 10% of the phenotypic variation for protein and sucrose content in soybean seed within the study population. Soybean breeding programs can use the results to improve soybean seed quality.

## 1. Introduction

An expanding market for soy foods in the U.S. makes selecting corresponding traits increasingly important for breeding programs. Large- and small-seeded soybean varieties are ideal for producing various soy foods, and high protein and sugar content are generally desired regardless of seed size [1]. Knowledge of correlations between agronomic and seed quality traits and the location of major and minor QTLs associated with these traits will assist plant breeders in improving the food quality characteristics of their germplasm.

Due to consumer demand for alternate plant-based protein, soy protein has the potential to become a valuable source of protein in the human diet. A review by Erdman and Fordyce (1989) [2] showed that soy protein contains an excellent amino acid profile. Soybeans also contain large amounts of other components of a complete diet, such as fiber and beneficial fatty acids. In addition, undesirable constituents, such as saturated fatty acids and cholesterol, are present in low amounts or are missing from soybean seed. Despite the high quality of soy protein, there has been a significant effort to improve the overall quality of protein content in soybean seed further. One example of progress in this area is the overexpression of the GmCGS2 gene in soybean seed to improve total protein content, including Met, a limiting amino acid in soybean [3]. Such breeding efforts will further enhance soybeans’ viability as a complete source of indispensable amino acids [4].

The average protein concentration for the 2023 U.S. soybean crop was 33.9%, 0.2 points lower than the average for the 2022 crop when analyzed on a 13% moisture basis. In 2015, 30% of samples with the highest and lowest protein concentrations were analyzed for their concentrations of amino acids. This analysis showed that samples with the highest protein concentrations were slightly lower in their concentrations of the five most limiting amino acids compared with the 30% of samples with the lowest total protein concentrations. This insight is helpful because soybeans with protein concentrations containing higher fractions of the most limiting amino acids are a positive indicator of the efficiency with which animals can successfully utilize soy-based protein in their diet [5]. In 2023, the same pattern was maintained. Lower crude protein samples showed a better relative abundance of the five most limiting amino acids than the higher crude protein samples [6]. Such findings justify research and breeding efforts to identify methods for improving amino acid profiles and total protein content in soybean seeds.

Protein content, like other traits in soybean, is quantitatively inherited. Because genotype-by-environment interactions influence quantitative characteristics, it is valuable to conduct genetic studies on protein content in multiple years and environments with different populations [7]. Environmental conditions such as water stress or high temperatures can cause protein content to decrease below desired levels [8]. The protein content of U.S. commodity soybeans typically ranges from 30 to 40%. However, it can vary more broadly from 25 to 50% Co-operative Feed Dealers, 2017). This trait exhibits high heritability (roughly 83 to 94%), and population means usually demonstrate additive effects, suggesting that additional variability can be incorporated from wild germplasm sources [9].

In another study, a soybean mapping population from a cross between Benning × Danbaekkong was grown in five environments. They reported four QTLs associated with protein and amino acids in chromosomes. 14, 15, 17, and 20 [10]. Wang et al. [11] developed two separate populations from two high-protein and two low-protein lines, R05-1415 × R05-638 and V97-1346 × R05-4256, which were grown in five different environments. They identified a major QTL associated with protein positioned on 20 across the environment and four other QTLs on chromosomes. 1, 5, and 14, which explained 10–16% of the total variation in protein content. A GWAS analysis of three hundred and eight recombinant inbred lines (RIL) revealed two major QTLs, *qPro15-1* and *qPro20-1*, associated with protein on chromosomes. 15 and 20, respectively [12]. Hyten et al. (2004) [13] identified four QTLs associated with protein content in 131 RILs, resulting in a cross between Essex and Williams. Chen et al. (2021) [14] conducted a study on a RILs population, resulting in a cross between the soybean genotypes Zhongdou27 and Hefeng25. They applied a high-density linkage map based on whole genome sequencing to detect QTLs associated with several agronomic traits and reported seven QTLs for protein content [15].

Specialty soybean food markets include tofu, natto, miso, and soymilk products. The presence and quantity of various sugars in raw soybean play a significant role in the taste and quality of these products. Miso and natto are two examples of soy food products that depend on the sugar profile of the soybean cultivar for sustainable production [16].

Of the approximately 33% of carbohydrates in soybean seed, roughly 16.6% can be classified as soluble sugars. The portion of sucrose over total sugars varies between 1.5 and 10.2% on a dry matter basis. Soluble sugars detected in soybean seed generally include sucrose, glucose, fructose, stachyose, and raffinose. However, the amounts vary significantly depending on the cultivar and maturity group under examination [16]. Stachyose and raffinose content ranged from 0.1 for stachyose to 2.1% for raffinose and 0.5 to 4.5% for stachyose dryly. Hou et al. [16] showed a distribution of sugar profiles into three categories. The first and most common category, or regular sugar profile, contained sucrose and stachyose as their main constituents. The second category identified represented a novel sugar profile with higher levels of both glucose and fructose. Such a profile would be valuable in breeding efforts to improve the digestibility of food products made from soybean. The final category identified was considered rare and valuable in breeding efforts intended for the soy food market because seed stachyose content would be limited. Because stachyose is generally undesirable, it is important to note that the 14 low stachyose genotypes discovered could be used as sources of genetic variations for modifying the sugar profile of soybean seed. Openshaw and Hadley [17] studied segregating generations from two different populations to discover heritability estimates for both populations’ parents and progeny. Results showed means of 0.38 and 0.14 for F2 plants and 0.69 for F3 progeny, which supported their observations for the heritability of sugar concentration.

Hou et al.(2009) [16] discovered multiple correlations between the five main detectable sugars of soybean seed. Strong positive correlations were observed for stachyose and total sugar (r = 0.73), raffinose and total sugar (r = 0.64), raffinose and stachyose (r = 0.68), sucrose and raffinose (r = 0.66), sucrose and total sugar (r = 0.82), as well as glucose and fructose (r = 0.98). The negative correlations included glucose or fructose as compared with sucrose (r = −0.68), raffinose (r = −0.59), total sugar (r = −0.27 to −0.24), and stachyose (r = −0.63 to −0.59). Sucrose and stachyose were negatively correlated at (r = −0.68). It is worth noting that the positive correlation reported here contradicts results published by Hymowitz et al. (1972) [18]. Relationships between individual plants sampled in a genetic study can significantly differ in the correlations discovered among traits. Therefore, just as studies on the location of QTLs for various traits should be conducted on multiple populations in different years and environments, the identification of correlations between traits should be studied for populations representing a wide range of genetic backgrounds [1].

Fitch et al. (2022) [19] conducted a GWAS on 266 soybean lines and identified seven QTL-associated chromosomes. 1, 6, 8, 9, 10 and 14. Liu et al. (2023) [20] performed high-density genetic mapping on 158 F7 lines resulting from a cross between Hefeng 25 and Zhongdou 27 and detected 16 QTLs associated with sucrose content. Another study constructed a RILs line population via interspecific hybridization between a William 82 and a wild soybean genotype, PI 483460B, and used 3343 polymorphic SNP markers to identify QTLs associated with soybean oil, protein, and sugar content. This study detected five, nine, and four QTLs for protein, oil, and sucrose content [21]. Salari et al. [22] conducted a study to identify QTLs associated with sucrose, raffinose, and stachyose in a RILs population resulting from a cross between two soybean lines—IA3023 and LD02-4485. They performed genotypic mapping using more than 3000 SNP markers and reported three QTLs—one on chromosome. 1 responsible for 10% variance and two major QTLs on chromosome 3, each explaining 22% phenotypic variation. Lee et al. [23] performed a GWAS on a collection of 220 soybean accessions using over 21 thousand SNP markers and identified seven major SNPs associated with sucrose on chromosomes. 2, 5, 8, 9, 10, 13, 14, and 15. A fine-mapping study was performed to identify sucrose content in a mapping population consisting of 190 Segment Substitution Lines (CSSLs) and identified a major QTL in chromosome 20. They screened nine candidate genes in the QTL region and reported that one gene was closely related to sucrose content in leaves.

The objectives of this study are to identify genetic structure associated with high sucrose and protein content in soybean lines grown in two locations in Arkansas. The Fayetteville Arkansas region (36.0627° N, 94.1606° W) is in northwest Arkansas in the Ozark Mountain Range between the Springfield Plateau and the Boston Mountains. The other environment in this study is Stuttgart, Arkansas (34.4941° N, 91.5581° W) located in Arkansas County; Stuttgart is in the Arkansas Delta. The second objective of this study is to identify extreme genotypes and recombinant types for protein and sucrose content.

## 2. Results and Discussion

As mentioned previously, due to the issue resulting from herbicide damage on the F2 generation and its effect on the later generations, an effort was made to minimize the number of plots lost in subsequent generations by using only those lines with an adequate seed number while considering the number of locations and replications needed. However, the issues were observed within plots each year. To improve statistical power in detecting QTLs, the decision was made to reduce the number of environments in the study to four (Fayetteville year 1, Fayetteville year 2, Stuttgart year 1, and Stuttgart year 2). This strategy positively impacts our research to overcome the problem. The ANOVA analysis of protein in the F2:3 population showed a normal distribution with a slight skew to the left (Figure 1A), suggesting that recombinant types for protein can inherit higher protein content than either parent, suggesting additive effects for this trait. These results are supported by similar investigations such as Qi et al. [24], Zhang et al. [25], and Teng et al. [26].

Similarly, the ANOVA analysis of sucrose on the F2:5 population showed a normal distribution with a slight skew to the left (Figure 1B), suggesting that recombinant types for sucrose can inherit higher protein content than their parents, implying an additive effect for this agronomic trait (Table 1).

In this study we have four environments (Fayetteville year 1, Fayetteville year 2, Stuttgart year 1, Stuttgart year 2). In these four environments, there are two important parameters (location and year). We decided to investigate the impact of these two parameters on sucrose and protein content. Analysis showed that there is a significant effect of location on protein content (Table 2). Despite no meaningful interaction of year on protein content, there is a significant interaction between year and location on protein which can be attributed to the location. Further statistical analysis showed there is a significant effect of year on sucrose content. While there was no meaningful interaction of location and sucrose content, there was significant interaction between year and location which can be attributed to the year effect. Single linear model analysis showed there is significant but negative correlation between protein and sucrose content with covariance = −0.81658 (Table 2). This result is supported by previous studies on protein and sugar correlations by both Maughan et al. [27] and Hymowitz et al. [18].

### 2.1. QTLs Associated with Protein

Two significant QTLs for protein content were located in this study. One QTL, observed across multiple locations and years, was situated on Chromosome. 20 between 2.28 × 10^7^ and 4.63 × 10^7^ bp, explaining 10% of the variation in protein content. This QTL was located between two SNP markers, Gm20_22826023_G_A and Gm20_46314790_C_T. In addition, this QTL co-localized with the previously reported QTL cqPRO-003 associated with protein [28]. The second QTL was located on chromosome. 11 between 3.77 × 10^7^ and 3.91 × 10^7^ bp and explained 9% of the variation for protein content. The QTL was flanked by Gm11_37749863_G_T and Gm11_39108822_A_G. In addition, both QTLs displayed a Linkage of Disequilibrium (LOD) value more significant than the threshold of 3 and additive effects of 0.5 and 0.4, respectively (Table 2, Appendix A). Overall, 241 QTLs through bi-parental study and 8 QTLs by GWAS study were reported in SoyBase [29]. Of these 249 reported QTLs, 25 were identified on chromosome 20, and 8 QTLs were identified on chromosome 11.

### 2.2. QTLs Associated with Sucrose

Two QTLs for sucrose content were located: one on chromosome 8 (3.804 × 10^7^ and 3.91 × 10^7^ bp) located between SNP markers Gm08_38044939_A_G and Gm08_39164008_C_T and the other one on chromosome. 14 (5.42 × 10^6^ and 7.85 × 10^6^ bp), positioned between Gm14_5418596_A_G and Gm14_7853431_G_T, which explained 8 and 11% of the variation for sucrose content, respectively. The LOD values of the QTLs on chromosomes 8 and 14 were 3.3 and 4.2, with additive effects of −0.3 and 1.1, respectively (Table 3, Appendix A). Overall, 37 QTLs were identified through bi-parental study and 2 by GWAS for sucrose content. Of these reported QTLs, only 1 was reported on Chromosome 14 [29].

In summary, protein and sucrose are two important soybean traits for different soy products. Along with increasing seed yield, improving protein and sucrose content are two goals in soybean breeding. Thus, many researchers have tried identifying genetic sources associated with these two agronomic traits. This research describes four QTLs in different chromosomes: two for total protein and two for percent sucrose. These QTLs offer a possible mode of selection for both traits separately and possibly simultaneously (when considering the QTLs on chromosome 11). We also report that the analysis of variance showed a significant GXE effect for protein content. There was also a strong environmental effect on protein and sucrose content. Therefore, field management, such as applying nitrogen, water availability, temperature, field location, etc., is essential for improving protein content [30,31].

To better understand how environmental factors influence protein and sucrose content, further investigation is needed to identify genetic correlations to specific soybean growing areas. These QTLs and associated markers could be potentially used for MAS on soybean seed sucrose content. As shown (Figure 1), there is some skew to the right for each graph, suggesting an additive effect and the possibility of selecting advantageous recombinant types for protein and sucrose content. Indeed, a Best Linear Unbiased Predictor (BLUP) analysis revealed UARK-2114, UARK-2116, and UARK-2141 as top performers for total seed protein and sucrose content in all environments.

## 3. Materials and Methods

### 3.1. Population Development and Field Experiment

The genetic population in this study was derived from a cross between high protein (R08-3221) and high sucrose (R07-2000) parents and, thus, represents a wide range of genetic diversity for seed protein and sucrose content. The high protein variety R08-3221 has an average of 47.9% protein, while the high sucrose parent R07-2000 averages 8.4 sucrose/stachyose. As a result, 186 true F1 hybrids were identified. The F2 population, consisting of 186 rows, was grown in Fayetteville, Arkansas. One single plant from each row was harvested to develop a mapping population. At the same time, the rest of each row was bulk harvested.

It should be noted that, unfortunately, due to herbicide damage, seed yield was low, and the plants produced thin pods with shriveled seeds. Although every line that produced a minimal amount of seed was harvested, those lines that displayed herbicide damage in the F2 generation also showed carry-over symptoms in subsequent generations. A quantity of 186 F2:3 populations from the individually harvested plants were advanced in the Costa Rica winter nursery. Then, a single plant from each F2:3 line was randomly selected, harvested, and advanced in the F3:4 generation the following summer. In addition, the remaining rows grown in Costa Rica were bulk harvested and grown as F2:5 lines in two Arkansas locations: Fayetteville (36.0627° N, 94.1606° W) and Stuttgart (34.4941° N, 91.5581° W) in a randomized complete block design with two replications in 3 m plots and 0.76 m row spacing.

### 3.2. Gentoyping

#### 3.2.1. DNA Extraction

The DNA was isolated from fresh and fully developed trifoliate leaves from each F2:4 generation using the CTAB protocol described by Doyle [32] with some modifications. Leaves were ground to a powder in liquid nitrogen and placed in 2 mL tubes. A total of 750 uL of an extraction buffer (2% CTAB, 100 mM Tris-Cl, 20 mM EDTA pH 8.0, 1.4 M NaCl, and 1% of volume β-mercaptoethanol) was added to each tube. Next, a water bath was used to incubate the tubes for 1 h at 60 °C, and then 1 mL chloroform: isoamyl alcohol (24:1) was added. The tubes were centrifuged at R.T. for 15 min at 12,000 rpm to separate the layers. The top layer was pipetted to a new tube containing 1 mL of ice-cold ethanol and mixed gently. DNA pellets were separated from the solution in a centrifuge at 14,000 rpm for 10 min at R.T. The excess liquid was discarded, and the pellets were washed in 75% ethanol and dried overnight. Finally, 200 uL of distilled sterile water was used to dissolve the DNA pellets. DNA concentration and quality were verified with NanoDrop^TM^ ND-2000 (Thermo Scientific, Waltham, MA, USA).

#### 3.2.2. Linkage Mapping and QTL Identification

For genetic mapping, DNA samples from the F2:4 generation and their parental lines were sent for genotyping using the BARC MSU Soy6k Illumina Infinium Genotyping HD Beadchip (652 K) on Illumina iScan (Illumina, San Diego, CA, USA) at the Michigan State University genotyping core facility, East Lansing, Michigan. Data from the analysis were then combined with the phenotypic data for protein content from the F2:4 and F2:5 generations. A total of 1922 polymorphic markers from the total of 6000 SNP markers from the 6K SNP chip were used for linkage mapping and then QTL analysis (Appendix A). MapDisto version 2.0 was then used to visualize the data and construct a linkage map. The Kosambi mapping function was used, and the Rf estimate was set as classical. The population type was a recombinant inbred line population to the fourth generation. The ordering criteria was SARF, and the ordering method was seriation. QGene version 4.4.0 was used to visualize those QTLs significantly affecting protein and sucrose content in each environment. Each of the environments was visualized separately and together. All of the QTL curves were overlaid to allow a straightforward interpretation of the effect of year and environment on the QTL detection (Appendix A).

### 3.3. Protein Analysis

Total Protein content was quantified using the DA 7250 NIR (near-infrared) analyzer instrument (Perten Instruments, Hägersten, Sweden) during the summer of 2016. Each line was measured non-destructively, as outlined in the user manual. However, many of the lines did not have enough seed to fill the cups evenly to the top, as recommended by the manufacturer. Therefore, a subsampling strategy was employed where a smaller cup held approximately 3 g of seed. Three subsamples were measured from each line and averaged to ensure that the smaller seed samples would provide a sufficiently accurate average compared with the larger sample size. No significant differences were observed between the subsampling and regular sampling methods. Instrument calibration files were built from the protein measurements of many yellow-seeded soybean varieties grown throughout the United States (Appendix A).

### 3.4. Sucrose Analysis

Total sucrose content was quantified via HPLC in collaboration with the National Center for Soybean Biotechnology and the Division of Plant Sciences at the University of Missouri. The system used was an Agilent 1200 series (Agilent USA) and was equipped with an evaporative light scattering detector (ELSD) to quantify the amounts of glucose, fructose, sucrose, raffinose, and stachyose contained in each of the samples. Samples were prepared for quantification by grinding and sifting the powder through 20-mesh sieves. The ground powder was then dried and lyophilized for 48 h, after which, 90.25 mg was weighed and transferred to fresh 2 mL tubes. The samples were then sent for HPLC analysis. At the University of Missouri, 900 uL of HPLC-grade water was added to each vial. The vials were agitated at 250 rpm for 30 min and then vortexed for 30 s. Once cooled, 900 uL acetonitrile was blended in, and the suspension was centrifuged for 30 min at 13.3 × 100 min^−1^× *g*. Five additional dilutions were performed on the supernatant with water: acetonitrile mixture of 65:35 (*v*/*v*). Sugar standards of 50, 100, 200, 300, 400, 500, and 1000 ug/mL were used for the HPLC analysis.

During analysis, the flow rate and gradient of two mobile phases were optimized to separate all sugars detected rapidly. Mobile phase A was pure water, while mobile phase B was acetonitrile: acetone mixture of 75:25 (*v*/*v*). The sample volume was 5 uL with high-grade nitrogen as the nebulizing gas at 3.4 bar. The column temperature was held constant at 35 °C, and the detector temperature was 55 °C [33] (Appendix A).

### 3.5. Data Analysis

For QTL analysis, a Best Linear Unbiased Predictor (BLUP) analysis was performed on the phenotypic data from the four population environments (Fayetteville year 1, Fayetteville year 2, Stuttgart year 1, Stuttgart year 2). The analysis was performed using the statistical analysis programming language and environment R (R: A language and Environment for Statistical Computing, 2021). This analysis revealed several possible recombinant types (UARK-2114, UARK-2116, and UARK-2141) high in total protein and sucrose content. Therefore, given the value of these traits for soy foods and feed, these lines could be used to develop soybean varieties high in total protein and sucrose content.

## 4. Conclusions

Protein and sugar content are important soybean agronomic characteristics used by food industries. Plant breeders aim to increase the value of their new cultivars in markets, especially for food and feeding purposes. We conducted this study in two years and two locations with a total of four environments to identify the QTLs associated with sucrose and protein. In this study, we identified four QTLs: two for protein located chromosomes 11 and 20 and two for sugar on chromosomes 11 and 14. The two detected QTLs on chromosome 11 were co-localized. This study can benefit plant breeders and the soybean market in general by accelerating the breeding process via marker-assisted selection.

## Figures and Tables

**Figure 1 plants-13-00650-f001:**
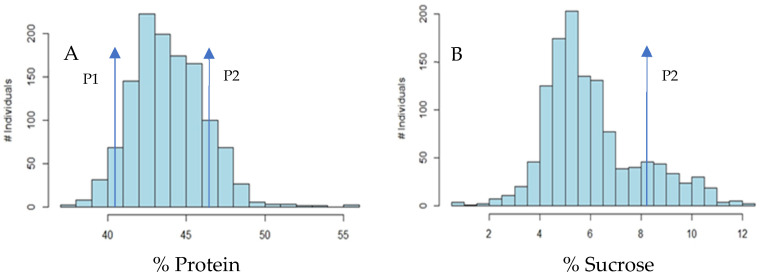
Histogram of percent seed protein content (**A**) and sucrose content (**B**). The population distribu-tion in (**A**), showing a normal distribution with a slight skew to the left, suggests that recombinant types for protein can inherit higher protein content than either parent, suggesting additive effects for this trait. The population distribution in (**A**), suggesting the skew of this trait to the right of the curve, could suggest additive effects for this trait, suggesting selection for advantageous recombinant types is possible. P1: Parental line R07-2000. P2: Parental line R08-3221, #: Symbol used as number.

**Table 1 plants-13-00650-t001:** ANOVA analysis of the environment in parental lines F2:5 population.

Trait	Parents		F2:5				
	R08-3221	R07-2000	µ *	Range ^†^	SD ^‡^	SE ^§^	F
Protein	47.9%	-	43.9	37.1–55.9	2.3	0.17	5.707 **
Sucrose	-	8.4%	6.1	0.5–12.2	1.88	0.14	2.0803 **

* μ, Average is estimated from F2:5 population resulting from cross between R08-3221 × R07-2000. ^†^, Range is the lowest and highest value of the trait under evaluation from the F2:5 population. ^‡^, SD represents standard deviation estimated variation within F2:5 population. ^§^, SE represents standard error measured from F2:5 population. **, Effects with *p*-values < 0.001 given between two parental lines of R08-3221 and R07-2000.

**Table 2 plants-13-00650-t002:** Interaction between environmental conditions on protein and sucrose content.

Trait	Estimate *	Std Error ^†^	t Ratio ^‡^	Prob > |t| ^§^
Protein				
Location	0.3574979	0.069492	5.14	<0.0001 **
Year	−0.113405	0.069492	−1.63	0.1030
Location × Year	0.3237548	0.069492	4.66	<0.0001 **
Sucrose				
Location	−0.009291	0.054748	−0.17	0.8653
Year	0.1724154	0.054748	3.15	0.0017 **
Location × Year	0.2043796	0.054748	3.73	0.0002 **
Protein × Sucrose	−0.230864	0.036256	−6.37	<0.0001 **

*, Estimate: an estimate of the model coefficients; ^†^, Std Error: an estimated standard deviation of the population for each of the parameters; ^‡^, t Ratio: the ratio of the estimate to its standard error; ^§^ Prob > |t|: the effect of the *p* value between two parameters. **, Effects with *p*-values < 0.001.

**Table 3 plants-13-00650-t003:** Detected QTLs associated with protein and sucrose content.

QTL	Chromosome.	Left Marker	Right Marker	Position (bp)	LOD	Add.
Pro-UA11	11	Gm11_37749863_G_T	Gm11_39108822_A_G	3.77 × 10^7^–3.91 × 10^7^	3.5	0.4
Pro-UA20	20	Gm20_22826023_G_A	Gm20_46314790_C_T	2.28 × 10^7^–4.63 × 10^7^	3.65	0.5
Suc_UA1	11	Gm11_38424068_C_T	Gm11_39108822_A_G	3.84 × 10^7^–3.91 × 10^7^	3.3	−0.3
Suc2_UA2	14	Gm14_5418596_A_G	Gm14_7853431_G_T	3.804 × 10^7^–3.96 × 10^7^	4.2	1.1

## Data Availability

The data presented in this study are available on request from the corresponding author.

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
