# Peer review of "Identification of Quantitative Trait Loci (QTL) for Sucrose and Protein Content in Soybean Seed"

_plants, 2024, doi:10.3390/plants13050650_

Round 1

Reviewer 1 Report

Comments and Suggestions for Authors

In this manuscript (plants-2828816) entitled "Identification of Quantitative Trait Loci (QTL) for Sucrose and Protein Content in Soybean Seed" submitted to Plants, Daniel R. Jamison and colleagues have identified five QTLs for soybean seed protein and sugar content. Two QTLs for protein content were located on chromosomes 11 and 20, two QTLs associated with sucrose content on Chr. 11 and Chr. 14, and one 11 associated with protein and sugar, explaining 10% of the phenotypic variation for protein and sucrose content in soybean seed within the study population. This study could benefit plant breeders and the soybean market by accelerating the breeding process via marker-assisted selection. This research is interesting and convincing, but minor points need to be addressed to improve the quality of this manuscript.

1. Authors should consider to show soybean seed phenotypes in Figure 1 in the revised manuscript.

2. For the discussion section, QTLs identified in this study should be discussed with comparison with QTLs or genes identified in previous studies.

3, Some citations like citations 1 and 2 appeared in red color in the maintext, which should be modified in the revised manuscript.

4, The population of recombinant inbred lines (RILs) employed in this studies should be introduced in details in the revision.

Author Response

Dear reviewer:

We thank you for your comments. We implemented your suggestions and significantly revised the manuscript. Moreover, we added more information in the introduction and reached the number of required words more than 4000. Please find our response to your comments below:

1- Authors should consider to show soybean seed phenotypes in Figure 1 in the revised manuscript.

Thank you for this suggestion and we added seed phenotype data sets for seed protein and sucrose content in supplementary table 3.

2- For the discussion section, QTLs identified in this study should be discussed with comparison with QTLs or genes identified in previous studies.

- Thank you for this helpful suggestion. We have added information from Soybase.org on QTL reported for protein and sucrose content. However, it was very difficult for us to compare our results with previous studies since our data is based on base pair locations and those studies were reported based on centimorgan distances. We found a reported QTL colocalizing to one of the QTL detected in this study and mentioned in the text.

3- Some citations like citations 1 and 2 appeared in red color in the main text, which should be modified in the revised manuscript.

-We thank you for noticing this issue and have corrected the red color in the manuscript.

4- The population of recombinant inbred lines (RILs) employed in this studiy should be introduced in details in the revision.

-We thank you for this suggestion and have included a supplementary table containing sucrose and protein measurements for all individuals in the population.

Reviewer 2 Report

Comments and Suggestions for Authors

Quality traits are becoming increasingly important in current plant breeding programs. This study addresses the possibility of using marker-assisted selection for important seed quality traits in soybean. The authors performed QTL mapping, for protein and sugar content by using a population of recombinant inbred lines (RILs) derived from a cross between a high protein and low sucrose line (R08-3221) and a high sucrose and low protein line (R07-2000). They identified five QTLs, two for protein content (chrom. 11 and 20), two for high sucrose content (Chr. 11 and Chr. 14), and one associated with both protein and sugar production, explaining 10% of the phenotypic variation of the traits. The subject matter falls within the scope of "Plants" journal. The paper shows an original contribution in the area of soybean molecular genetics with respect to the seed quality. The methodology is appropriate but some additions and clarifications are required through the text.

Hereafter, some specific comments for authors to consider in their revision.

- There is no indication on how many molecular markers were generated and used for the linkage analysis and to detect the QTLs, or no enough bibliographic reference on this point. No molecular details for the presented markers are given. I think readers of this article would be interested to get this information.

- In my opinion the most interesting part of this work is the assessment of the traits over 2 consecutive years and different locations. This provides the possibility to the authors to make valuable statement about QTL stability over the 2 years and different environments. Which of the detected QTL are "location" or "year" specific? This should be shown in the table. The QTL* year or QTL * environment interaction might appeal to a broader readership.

- Figure 1. The parental line R08-3221 is indicated as the low-protein content parent (figure 1a). This is not consistent with the text and the Table-1. The P1 is not indicated in the histogram showing the % of sucrose (figure 1b).

- Line 118-119. "However, the issues were observed within plots each year". This sentence is not clear.  What does the word "issues" refer to? This point should be revised.

- It is not clear if the ANOVA analysis on sugar content performed on a F2:3 or a F2:5 population as stated in Table 1 legends. This should be clarified.

- Line 164. "The LOD values of QTLs on Chr. 8 and 14 were 3.3 and 4.2, with additive ...".  The Chr. 14 is not stated in the table 2. This should be corrected.

- Table 2 ... The markers Gm08_39164008_C_T2 is present in chromosome 8 and 11. How possible is that? Similarly, the marker Gm11_39108822_A_G is mapped in two locations in chr. 11. This may point to large number of missing values (technical problems encountered during analysis) or considerable genotypic errors. No explanation is provided. The authors should clarify this. It would be easier if the authors graphically illustrate the linkage map around the detected QTLs.

- Lines 178-179. "We also report that the analysis of variance showed a significant GXE effect for protein content. There was also a strong environmental effect on protein and sucrose content". This is not indicated (or clearly shown) in the results section in figures or tables.

Line 238-239. "Each of the environments was visualized separately and together.  All of the QTL curves were overlaid to allow a straightforward interpretation of the effect of year and environment on the QTL detection. This is again not shown in the results section or presented into the figures/Tables. The authors should show/publish the QTL curves that are claimed.

Line 108., "The objectives of this study are to find novel, independent QTL and to confirm..." What does it mean independent QTL? Please rephrase.

Comments on the Quality of English Language

The English language is good. Minor editing is required. The authors should improve the clarity in some sentences (like, Lines 118, 115-125). I have included this in the comments.

Author Response

Dear reviewer:

We appreciate you for your comments. We implemented your suggestions and significantly revised the manuscript. As you recommended, We added tables and supplementary data to support our study. Please find our response to your comments below:

1- There is no indication of how many molecular markers were generated and used for the linkage analysis and to detect the QTLs or not enough bibliographic reference. No molecular details for the presented markers are given. I think readers of this article would be interested to get this information.

Thank you for your interest in the molecular marker data and the suggestion of including more information on this point. Of the 4,099 heterozygous markers, 1,374 were used in the Qgene analysis. The total number of SNP Chip markers used in the QTL analysis was 1,922. We have added this information to the manuscript.

2- In my opinion, the most interesting part of this work is the assessment of the traits over 2 consecutive years and different locations. This allows the authors to makforvaluable statements about QTL statements over the 2 years and different environments. Which detected QTL are "location" or "year" specific? This should be shown in the table. The QTL* year or QTL * environment interaction might appeal to a broader readership.

This was a very helpful suggestion. We have added an analysis of the location by year effect in each of the four environments in this study. We have decided to focus on only those QTL that were significant in all four environments in this study. This gives us the best chance to identify more impactful QTL that will be informative for other breeding programs and research groups.

3- Figure 1. The parental line R08-3221 is indicated as the low-protein content parent (figure 1a). This is not consistent with the text and the Table-1. The P1 is not indicated in the histogram showing the % of sucrose (figure 1b).

Thank you for catching this. We had a labeling error. We fixed the P1 and P2 designations in the text and in the figures.

4- Line 118-119. "However, the issues were observed within plots each year". This sentence is not clear. What does the word "issues" refer to? This point should be revised.Rewrite this part of the paragraph to make the "issues" with plots more clearly stated.

Done

5- It is not clear if the ANOVA analysis on sugar content performed on a F2:3 or a F2:5 population as stated in Table 1 legends. This should be clarified.

Thank you for this suggestion. We looked at the ANOVA analysis and as far as we can tell it is stated in Table 1 that the ANOVA analysis was performed on an F2:5 population.

5- Line 164. "The LOD values of QTLs on Chr. 8 and 14 were 3.3 and 4.2, with additive ...". The Chr. 14 is not stated in the table 2. This should be corrected.

We appreciate you catching this issue. This information was likely cut off during formatting and we have corrected the issue.

7- Table 2 ... The markers Gm08_39164008_C_T2 is present in chromosome 8 and 11. How possible is that? Similarly, the marker Gm11_39108822_A_G is mapped in two locations in chr. 11. This may point to large number of missing values (technical problems encountered during analysis) or considerable genotypic errors. No explanation is provided. The authors should clarify this. It would be easier if the authors graphically illustrate the linkage map around the detected QTLs.

Thank you for catching and identifying this error. We have identified the issue with the markers and believe that we have resolved the issue. The table is revised, and the issue is resolved

8- Lines 178-179. "We also report that the analysis of variance showed a significant GXE effect for protein content. There was also a strong environmental effect on protein and sucrose content". This is not indicated (or clearly shown) in the results section in figures or tables.

Excellent suggestion. We performed ANOVA and provided the requested. I believe this information is interesting. We have added GXE to the ANOVA table 2.

9- Line 291-293. "Each of the environments was visualized separately and together. All of the QTL curves were overlaid to allow a straightforward interpretation of the effect of year and environment on the QTL detection. This is again not shown in the results section or presented into the figures/Tables. The authors should show/publish the QTL curves that are claimed.

Good point! We have compiled a supplementary table with all the plots of QTL identified for each trait. Those QTL being significant to all environments were plotted and included in the supplementary file.

10-Line 108., "The objectives of this study are to find novel, independent QTL and to confirm..." What does it mean independent QTL? Please rephrase.

- We appreciate this suggestion and have had some discussion on the objectives of this study and have re-written our objective statements to reflect our conclusions. Namely, to identify genetic structure associated with sucrose and protein content in soybean lines grown in Arkansas.

Round 2

Reviewer 1 Report

Comments and Suggestions for Authors

Authors have addressed my concerns in the revision.

Reviewer 2 Report

Comments and Suggestions for Authors

The authors have  addressed my concerns. The paper is ready now for publication.